# Effect of Potassium Promoter on the Performance of Nickel-Based Catalysts Supported on MnOₓ in Steam Reforming of Ethanol

Magdalena Greluk [1,*], Marek Rotko [1], Grzegorz Słowik [1], Sylwia Turczyniak-Surdacka [2], Gabriela Grzybek [3], Kinga Góra-Marek [3] and Andrzej Kotarba [3]

[1] Department of Chemical Technology, Faculty of Chemistry, Maria Curie-Sklodowska University in Lublin, Maria Curie-Sklodowska Sq. 3, 20-031 Lublin, Poland; marek.rotko@mail.umcs.pl (M.R.); grzegorz.slowik@mail.umcs.pl (G.S.)

[2] Biological and Chemical Research Centre, University of Warsaw, 101 Żwirki i Wigury Street, 20-089 Warsaw, Poland; sturczyniak@cnbc.uw.edu.pl

[3] Faculty of Chemistry, Jagiellonian University, Gronostajowa 2, 30-387 Krakow, Poland; g.grzybek@uj.edu.pl (G.G.); kinga.gora-marek@uj.edu.pl (K.G.-M.); ak@uj.edu.pl (A.K.)

[*] Correspondence: magdalena.greluk@mail.umcs.pl; Tel.: +48-81-537-55-14; Fax: +48-81-537-55-65

**Abstract:** The effect of a potassium promoter on the stability of and resistance to a carbon deposit formation on the Ni/MnOₓ catalyst under SRE conditions was studied at 420 °C for different $H_2O$/EtOH molar ratios in the range from 4/1 to 12/1. The catalysts were prepared by the impregnation method and characterized using several techniques to study their textural, structural, and redox properties before being tested in a SRE reaction. The catalytic tests indicated that the addition of a low amount of potassium (1.6 wt.%) allows a catalyst with high stability to be obtained, which was ascribed to high resistance to carbon formation. The restriction of the amount of carbon deposits originates from the potassium presence on the Ni surface, which leads to (i) a decrease in the number of active sites available for methane decomposition and (ii) an increase in the rate of the steam gasification of carbon formed during SRE reactions.

**Keywords:** hydrogen production; ethanol steam reforming; nickel-based catalyst; manganese oxides; potassium promoter

## 1. Introduction

Hydrogen does not affect the environment adversely, unlike fossil fuels and nuclear power; thus, it is considered a clean next-generation energy source with great potential for commercialization. Unfortunately, it is not freely available in nature and it must be produced. For hydrogen to become a truly sustainable energy source its production from renewable resources should be promoted. Ethanol is an attractive choice of raw material for producing hydrogen because, besides its low ecotoxicity, it can be produced from biomass and utilized without the net addition of carbon dioxide to the atmosphere. Ethanol produced by biomass fermentation has a high ratio of water to ethanol, which is suitable for the steam reforming reaction. In the case of this process, it is not necessary to carry out the distillation of fermented to obtain pure ethanol. Therefore, it is possible to directly use bioethanol obtained from biomass fermentation in steam reforming reactions, which can save lots of money and energy for purification [1–4].

One of the greatest challenges in the production of renewable hydrogen from bioethanol using the catalytic steam reforming of ethanol (SRE) process is the development of a catalyst with the desirable activity, selectivity, and stability. Transition metals have widespread applications as heterogeneous catalysts for a number of reactions and two of them, i.e., cobalt and nickel, possess an excellent performance in the SRE reaction due to their high C–C bond cleavage activity. The interaction of a transition metal with a support has a

significant influence on the structure definition and the catalytic activity in the SRE process. Supporting metals might allow the production very active nanoparticles stabilized towards sintering and somehow activated by the interaction with the support. The most often used supports for cobalt- or nickel-containing catalysts in the SRE process are oxides [5–8]. Alumina is one of the most widely used supports in catalysis due to its high surface area, which allows a great dispersion of the active phase [3], and its application as the support of cobalt- or nickel-based catalysts in the SRE process has been extensively studied [9–14]. However, the acidic character of alumina may promote ethanol dehydration to ethylene, which can lead to carbon formation on a catalyst surface. On the other hand, redox supports like ceria and ceria-containing mixed oxides have been proposed to prevent the carbon deposits on the catalytic structure under SRE conditions, due to their high oxygen storage capacity [7,15]. Although the previously mentioned redox support oxides are highly favorable for ethanol reforming, the unique redox property and strong oxygen storage and release ability of manganese oxides ($MnO_x$) were practically not investigated in this process. It is all the more remarkable that manganese oxide-based systems represent a very intriguing and peculiar class of support with both high chemical stability and the enhanced redox behavior, allowing the stabilization of various intermediate manganese oxidation states ($MnO_2$, $Mn_5O_8$, $Mn_2O_3$, $Mn_3O_4$, and $MnO$) and crystalline phases, which provides potential suitability for catalytic reactions [8,16,17].

In the literature, there are only a few studies on the application of manganese oxides in an SRE reaction over cobalt- or nickel-based catalysts. Most of these reports concern the application of this oxide as a modifier of cobalt-based catalysts [5,18,19]. Lee et al. [18] showed that the introduction of manganese oxides to the $Co_{10}Si_{90}MCM$-48 material had a favorable effect on its catalytic performance in the SRE process by preventing the catalytic deactivation caused by the sintering of cobalt particles. Moreover, according to the same authors [18], manganese oxides appeared to provide oxygen to cobalt species, resulting in increases in hydrogen production and the suppression of carbon monoxide production. The same conclusions were drawn by Kwak et al. [19] for the role of manganese oxide in the spinel structured $Co_2MnO_4$ of $Co_2MnO_4$/SBA-15 material during its catalytic performance in the SRE reaction. However, not only positive effects on the performance of manganese-oxide-modified cobalt-based catalysts under the SRE conditions were observed. As was presented by Li et al. [5], an addition of manganese oxides to the $Co/Al_2O_3$ catalyst suppressed its activity in the SRE process. Furthermore, a few reports concerning the application of manganese oxide as a support of cobalt- and nickel-based catalysts can be found in the literature [2,20,21]. Both cobalt- and nickel-based catalysts supported on two structured manganese oxides, i.e., birnessite and todorokite, were found to be highly active and selective to produce hydrogen in the SRE reaction [2,20]. In addition, the cryptomelane-based cobalt-manganese catalyst showed high ethanol conversion and selectivity to hydrogen and carbon dioxide in the SRE reaction in the temperature range of 390–480 °C. However, it was observed only at the initial stage of the process. With the increase in the time-on-stream, its activity decreased with a simultaneous increase in the acetaldehyde production, which was ascribed to deactivation due to carbon formation [21]. Sohrabi and Irankhah [22] showed that $Ni/CeMnO_2$ catalyst was active in the SRE process and its activity was improved by the addition of copper or iron promoters.

Carbon deposition is the major deactivation mechanism of the ethanol steam reforming catalysts. Although reducibility and oxygen transfer capacity of support have shown to be fundamental in the keeping the active phase surface free of carbon deposits [3], the addition of alkali is another solution that is also commonly used to reduce the problem of carbon formation under the SRE conditions [22–33]. In general, the application of alkali promoters allows a decrease in the rate of formation of carbon deposits on the surface of the catalysts during the SRE process [25,31–34]. However, Grzybek et al. [29] indicated that the rate of carbon depositions on the surface of a potassium-doped CoZn|a-$Al_2O_3$ catalyst is even higher compared to the undoped sample and its enhanced stability under SRE conditions results from a stronger interaction between the cobalt nanocrystals and

the alumina support in the presence of a potassium promoter, stabilizing the active phase against the driven detachment and its subsequent encapsulation by the growing carbon deposit. Moreover, Słowik et al. [28] showed that the presence of a potassium promoter does not protect the $Ni/CeO_2$ catalyst against carbon deposit formation and even worsens its stability under SRE conditions compared to the unpromoted sample. According to the authors, the addition of a potassium promoter influences the structure and degree of graphitization of the formed carbon deposits, leading to its faster deactivation.

Considering the above points, the purpose of the present work was to study the performance of the $Ni/MnO_x$ and $KNi/MnO_x$ catalysts under an SRE reaction and to determine the role of a potassium promoter on the stability and resistance to carbon deposit formation. The catalysts were prepared by the impregnation method and characterized using several techniques in order to study their textural, structural, and redox properties before being tested in an SRE reaction.

## 2. Experimental Procedure

### 2.1. Catalyst Synthesis

$(CH_3COO)_2Mn\cdot4 H_2O$ ($\geq$97.0%, Sigma-Aldrich, Darmstadt, Germany), $Ni(NO_3)_2\cdot6 H_2O$ (>97.0%, Sigma-Aldrich, Darmstadt, Germany), $KNO_3$ ($\geq$99.0%, Merck, Darmstadt, Germany), and $(NH_4)_2CO_3$ (100%, Avantor, Gliwice, Poland) were used without further purification.

In the first step, the $MnO_x$ support was prepared using the conventional precipitation method from a 0.5 mol/L manganese acetate aqueous solution. Precipitation was accomplished at 40 °C with the addition of 1 mol/L ammonium carbonate solution, drop by drop, up to a pH of 8, under continuous stirring in suspension. After the aging of the precipitate at 60 °C for 2 h, the suspension was filtrated. The filtrate was washed with absolute ethanol in order to remove water from the precipitates. The obtained solid was dried overnight at 110 °C and then calcined at 500 °C in air.

In the second step, the $Ni/MnO_x$ catalysts were prepared using the wet impregnation method, with 10 g of a finely grinded $MnO_x$ solid and solutions containing 5.505 g of nickel nitrate. The mixture was dried at 110 °C for 12 h and then calcined at 500 °C in air to decompose the nickel nitrate into the nickel oxide precursor. The loading of cobalt or nickel to $MnO_x$ was 10 wt%.

In the final step, the $KNi/MnO_x$ catalysts were prepared using the wet impregnation method, with 5 g of a finely grinded $Ni/MnO_x$ solid and solutions containing 0.264 g of potassium nitrate. The mixture was dried at 110 °C for 12 h and then calcined at 500 °C in air to decompose the potassium nitrate. The loading of potassium to $Ni/MnO_x$ was 2 wt%.

### 2.2. Catalysts Characterization

The chemical composition of the cobalt- and nickel-based catalysts was determined by the X-ray fluorescence method, using an Axios mAX (PANalytical, Malvern, UK) fluorescence spectrometer. The textural properties of the support and catalysts were determined from nitrogen adsorption-desorption isotherms at −196 °C using ASAP 2405N (Micromeritics, Norcross, GA, USA). The specific area was obtained using the BET method and pore volume and diameter were obtained using the BJH method. The linear region in the BET plots is between $p/p^0 = 0$ and 0.25. Prior to the analysis, the samples were outgassed at 200 °C. The X-ray diffraction (XRD) patterns were collected with an Empyrean X-ray (PANalytical, Malvern, UK) diffractometer with Cu K$\alpha$ radiation ($\lambda$ = 0.154 nm) in the 20–100° 2$\theta$ range and a step of 0.026°. In order to analyze samples in their reduced form, prior to the analysis, they were activated in situ at 500 °C with hydrogen in an XRK 900 reactor chamber (Anton Paar, St. Albans, UK). The morphology and lattice profiles of all the catalysts after their reduction were characterized by an electron transmission microscope Titan G2 60–300 kV (FEI Company, Hillsboro, OR, USA). The fast Fourier transform (FFT) was obtained to determine which d-spacing corresponded to which crystalline species. The mapping was carried out in the STEM mode by collecting a point-by-point EDS spectrum of each of the corresponding pixels on the map. By measuring the size of at

least 200 particles from the HRTEM images, particle size distributions and mean particle size data were obtained. Prior to the analysis, the catalysts were reduced in a fixed-bed reactor with hydrogen at 500 °C and transferred in a closed reactor to a glove box filled with argon, which prevented the catalyst from oxidation. Then, a TEM vacuum transfer holder was used to transfer the catalysts laid on a copper grid in the glove box, from the aforementioned preparation station into a TEM microscope in argon atmosphere. SEM images of the catalysts used in the SRE process were taken using a FIB-SEM Crossbeam 540 FEG (Zeiss, Jena, Germany) with an acceleration voltage of 1.8 kV. The samples were prepared by drying the solution of catalyst diluted in ethanol dropped onto carbon tape. TEM images of catalysts used in the SRE process were obtained in an electron transmission microscope, Titan G2 60–300 kV (FEI Company). Samples were suspended in ethanol and exposed to ultrasonic vibrations to decrease aggregation. A drop of the resultant mixture was laid on the copper grid covered with lacey formvar and stabilized with carbon.

$H_2$-TPR measurements were carried out with an AutoChem II 2920 (Micromeritics, Norcross, GA, USA) analyzer. Typically, a 50 mg sample was placed in a quartz U-tube reactor and heated to 750 °C at 10 °C/min under the mixed reduction gas flow (5% $H_2$/He, 30 mL·min$^{-1}$). The hydrogen consumption was monitored by a thermal conductivity detector (TCD).

The acidity of the samples was determined by the FT-IR pyridine adsorption studies. The sample was pressed into the form of a disc, then placed in a custom-made quartz IR cell and in situ evacuated, then activated at 500 °C in a pure hydrogen stream for 1 h.

To estimate the impact of potassium on the catalyst's electron-donor properties, the work function was determined. The contact potential difference (VCPD) was measured by means of the dynamic condenser method of Kelvin using a KP6500 probe (McAllister Technical Services, Berkeley, CA, USA). A stainless-steel plate was used as a reference electrode ($\phi_{ref}$ = 4.3 eV) at 3 mm in diameter. The measurement parameters were: the vibration frequency at 120 Hz and the amplitude at 40 a.u. The measurements were performed at ambient conditions (room temperature, atmospheric pressure). The work function values were calculated based on the equation: $V_{CPD} = \phi_{ref} - \phi_{sample}$.

X-ray photoelectron spectroscopy (XPS) studies were performed in a Kratos Axis Supra Spectrometer equipped with a monochromatized Al source operating at 150 W over the samples mounted on conductive double-side Cu tape. The pass energy of the analyzer was set to 160 eV (energy step 1.0 eV) for the survey scan and 20 eV (energy step 0.1 eV) for high-resolution Ni 2p, Mn 2p, O 1s, C 1s, and K 2p spectra of spent catalysts (EtOH/$H_2O$ = 1/12, 420 °C). The base pressure in the analysis chamber was $4 \times 10^{-7}$ Pa. Data processing was performed with CasaXPS software (v 2.3.16 PR 1.6, Zurich, Switzerland), taking into account the relative sensitivity factors (provided by CasaXPS software). The XPS spectra were charge-corrected for an O 1 s peak binding energy equal to 530.0 eV. After conducting Shirley background subtraction, the quantification of the Ni 2p and Mn 2p regions was performed using the peak fitting procedure of mixed Gaussian–Lorentzian components, except for the metallic cobalt and nickel for fitting, of which the LA(1.4,5,5) and LA(1.1,2.2,10) mixed functions were used, respectively. The fitting parameters for those regions were constrained and determined from the reference spectra recorded over: Ni(0) from metallic nickel and Ni(II) from nickel(II) oxide.

Thermogravimetric (TG) analysis was conducted on a TG121 microbalance system (CAHN, Newington, NH, USA) in order to quantify the amount of carbon deposited onto the catalyst surface under SRE conditions. The studies were carried out at the temperature of 420 °C at two steam to ethanol ratios, i.e., $H_2O$:EtOH = 4:1 and 12:1. The total volumetric flow rate of the mixture (70 mL min$^{-1}$) was kept constant by adding helium. Prior to reaction, the catalyst sample (0.01 g) was reduced by passing 10% $H_2$/He flow at the temperature of 500 °C for 1 h.

### 2.3. Catalyst Evaluation in the SRE Process

Studies were performed using the application of a Microactivity Reference unit (PID Eng & Tech., Madrid, Spain), similar to that described in our earlier reports. The samples of catalysts diluted with quartz were introduced to the fixed-bed continuous-flow quartz reactor. Activation was performed in a stream of hydrogen with a flow rate of 100 mL min$^{-1}$ at 500 °C under isothermal conditions for 1 h. After activation, the catalysts were cooled down to 420 °C, and then hydrogen was replaced by the reaction mixture composed of $H_2O$/EtOH/Ar = 48/4/52 vol%, $H_2O$/EtOH/Ar = 36/4/60 vol%, $H_2O$/EtOH/Ar = 24/4/72 vol%, and $H_2O$/EtOH/Ar = 16/4/80 vol% for $H_2O$/EtOH molar ratios of 12/1, 9/1, 6/1, and 4/1, respectively. The total flow rate was equal to 100 mL min$^{-1}$ and the weight of the catalyst was equal to 100 mg. Space velocity referenced to the total flow rate of the reaction mixture divided by the catalyst weight was equal to 60,000 mL h$^{-1}$ g$^{-1}$ (space velocity referenced to the flow rate of EtOH was regarded as a key component of the reaction; divided by the catalyst weight, it was equal to 2400 mL $_{EtOH}$ h$^{-1}$ g$^{-1}$). The reaction substrates and products were analyzed with two online gas chromatographs. One of them, Bruker 450-GC was equipped with two columns, the first filled with a porous polymer Porapak Q (for all organics, $CO_2$ and $H_2O$ vapor) and the other a capillary column CP-Molsieve 5Å (for $CH_4$ and CO analysis). Helium was used as a carrier gas and a TCD detector was employed. The hydrogen concentration was analyzed by the second gas chromatograph, Bruker 430-GC, using a Molsieve 5Å, argon as a carrier gas, and a TCD detector. Both Bruker 450-GC and Bruker 430-GC were equipped with TCD detectors.

The conversion of ethanol ($X_{EtOH}$) and conversions of ethanol into individual carbon-containing products ($X_{CP}$) were calculated on the basis of its concentrations at the reactor inlet and outlet:

$$X_{EtOH} = \frac{C_{EtOH}^{in} - C_{EtOH}^{out}}{C_{EtOH}^{in}} \times 100\% \tag{1}$$

$$X_{CP} = \frac{n_i C_i^{out}}{\sum n_i C_i^{out}} \times 100\% \tag{2}$$

where $C_{EtOH}^{in}$-is the molar concentration of ethanol in the reaction mixture (mol%), $C_{EtOH}^{out}$-is the molar concentration of ethanol in the post-reaction mixture (mol%), $C_i^{out}$-is the molar concentration of carbon-containing product in the post-reaction mixture (mol%), and $n_i$–is the number of carbon atoms in the carbon-containing molecule of the reaction product.

The selectivity of hydrogen formation was determined as:

$$H_2 \text{selectivity} = \frac{C_{H_2}^{out}}{C_{H_2}^{out} + 2 \times C_{CH_4}^{out} + 2 \times C_{CH_3CHO}^{out}} \times 100\% \tag{3}$$

where $C^{out}$-is the molar concentration of the hydrogen-containing products in the post-reaction mixture (mol%).

## 3. Results and Discussion

### 3.1. The Influence of a Potassium Promoter on the Catalysts' Physicochemical Properties

The chemical compositions and textural properties of $MnO_x$ support and both nickel-based catalysts are summarized in Table 1. The results of XRF studies reveal that the assumed content of nickel is obtained with satisfactory accuracy. However, the actual weight percentage of potassium loadings in the $KNi/MnO_x$ catalyst is less than the nominal value. It is well-known that in order to achieve high metal dispersion and high thermal stability, the support material should have a high specific surface area [35,36]. However, the BET surface area of the obtained $MnO_x$ support was very low, which led to obtaining the catalysts with low surface areas and rather low nickel dispersion. TEM studies of the $Ni/MnO_x$ catalyst (Table 1, see Supporting Information, Figure S1a) demonstrate that, although the Ni$^0$ particles have a narrow size distribution in the range of 6–24 nm, the

average particle size is high and equal to 14.2 nm. Compared to the $Ni/MnO_x$ catalyst, $Ni^0$ particles of the $KNi/MnO_x$ sample (Table 1, see Supporting Information, Figure S2b) exhibited a wider size distribution in the range of 6–38 nm with a larger average particle size of 20.4 nm. Based on the Scherrer equation analysis of the (111) reflections, the average size of the obtained $Ni^0$ particles of the $Ni/MnO_x$ and $KNi/MnO_x$ catalysts was estimated to be 19.0 and 23.8 nm, respectively (Table 1). Although slightly different values of $Ni^0$ particle size were found for TEM and XRD methods, a trend of increasing $Ni^0$ particle size in the presence of a potassium promoter is the same. Probably, the aggregation of the primarily formed nickel particles of the $Ni/MnO_x$ sample during its second calcination after impregnation with potassium salt led to the formations of nickel particles with a larger size.

**Table 1.** Physical and chemical properties of the $MnO_x$ support and nickel-based catalysts.

| Sample | Metal Content (wt.%) | | $S_{BET}$ $(m^2\ g^{-1})$ | $Ni^0$ Particle Size (nm) * | |
| --- | --- | --- | --- | --- | --- |
| | Ni | K | | By XRD | By TEM |
| Ni/MnO | 9.7 | - | 16.7 | 19 | 14 |
| KNi/MnO | 9.1 | 1.6 | 10.1 | 23 | 20 |
| $MnO_x$ | - | - | 12.0 | - | - |

* $Ni^0$ denoted Ni particle size for catalyst reduced at 500 °C.

HRTEM studies were carried out to obtain an overview of the morphology and latticed spacings of nickel-based catalysts after their reduction at 500 °C. A representative selection of HRTEM images of the $Ni/MnO_x$ and $KNi/MnO_x$ catalysts is shown in Figure 1a–b. The corresponding fast Fourier transform (FFT) pattern (Figure $1a_1,b_1$) adequately verifies the formation of metallic particles of nickel in its elemental state $Ni^0$ and low valent manganese oxide MnO after thermal treatment of both catalysts under a reducing atmosphere. The FFT patterns indicate the characteristic (111) intensity ring and plane, with a d-spacing of 0.203 nm for face-centered cubic metallic nickel. the d-spacing of 0.222 nm correspond to the (200) lattice fringes of cubic MnO. These findings are consistent with the results obtained by X-ray diffraction studies (Figure 2c,d, see Supporting Information, Figure S2b). Recorded XRD patterns of both catalysts as-prepared and reduced with hydrogen at 500 °C confirm the structural transformation that accompanies their reduction. In the case of as-prepared samples (Figure 2a,b), the diffraction peaks at 2θ exhibited a mixed-phase structure of NiO, $Mn_2O_3$, and $Mn_3O_4$; the XRD results of the reduced samples (Figure 2c,d) indicate that the series of Bragg reflections corresponding to two crystalline phases, $Ni^0$ and MnO, is maintained (see Supporting Information, Table S1) [37–42]. This means that, during the treatment of both nickel-based catalysts under a hydrogen atmosphere at 500 °C, $Ni^0$ and MnO phases are formed by the reduction of NiO and $Mn_2O_3/Mn_3O_4$ phases, respectively. The comparison of XRD results for the as-prepared $MnO_x$ support (see Supporting Information, Figure S2a) and both nickel-based catalysts (Figure 2a,b) indicates a change in the structure of the $MnO_x$ phase during the step of catalyst synthesis concerning nickel introduction. The as-prepared $MnO_x$ support mainly exhibits a phase of hausmannite ($Mn_3O_4$) with spinel structure and tetragonal symmetry, with a minor contribution of cubic $Mn_2O_3$ phase, while XRD patterns of both as-prepared catalysts include a dominant $Mn_2O_3$ phase and a minor phase of hausmannite ($Mn_3O_4$) [43]. This indicates that, during the calcination of the nickel phase precursor at the temperature of 500 °C, the partial transformation of $Mn_3O_4$ to $Mn_2O_3$ occurs, which is closely related to the oxygen transfer on the surface: $O_2$ (gas) $\leftrightarrow$ $O_2$ (ads) $\leftrightarrow$ $O_2^-$ (ads) $\leftrightarrow$ $2O^-$ (ads) $\leftrightarrow$ $2O^{2-}$ (ads) $\leftrightarrow$ $2O^{2-}$ (lattice) [44]. Because there is not much difference between the XRD patterns of the $Ni/MnO_x$ and $KNi/MnO_x$ catalysts, it can be suggested that no phase is changed after the potassium addition. Any phases containing potassium are not identified, indicating that the potassium compound remained amorphous [30] or was highly dispersed on the catalysts' surface [21]. STEM-EDS analysis (Figure $1b_3$) confirms that potassium is well-dispersed and distributed over both MnO and $Ni^0$ particles of the reduced $KNi/MnO_x$ sample. On the

other hand, the same analysis indicates an inhomogeneity of Ni$^0$ distribution on the MnO support for both the reduced Ni/MnO$_x$ (Figure 1a$_3$) and KNi/MnO$_x$ (Figure 1b$_3$) catalysts. From visual inspections of the maps, it is clear that in the case of both nickel-based catalysts, there are Ni-rich and Ni-poor regions but it is difficult to establish which nickel-based catalyst exhibited more uniformly distributed Ni$^0$ particles.

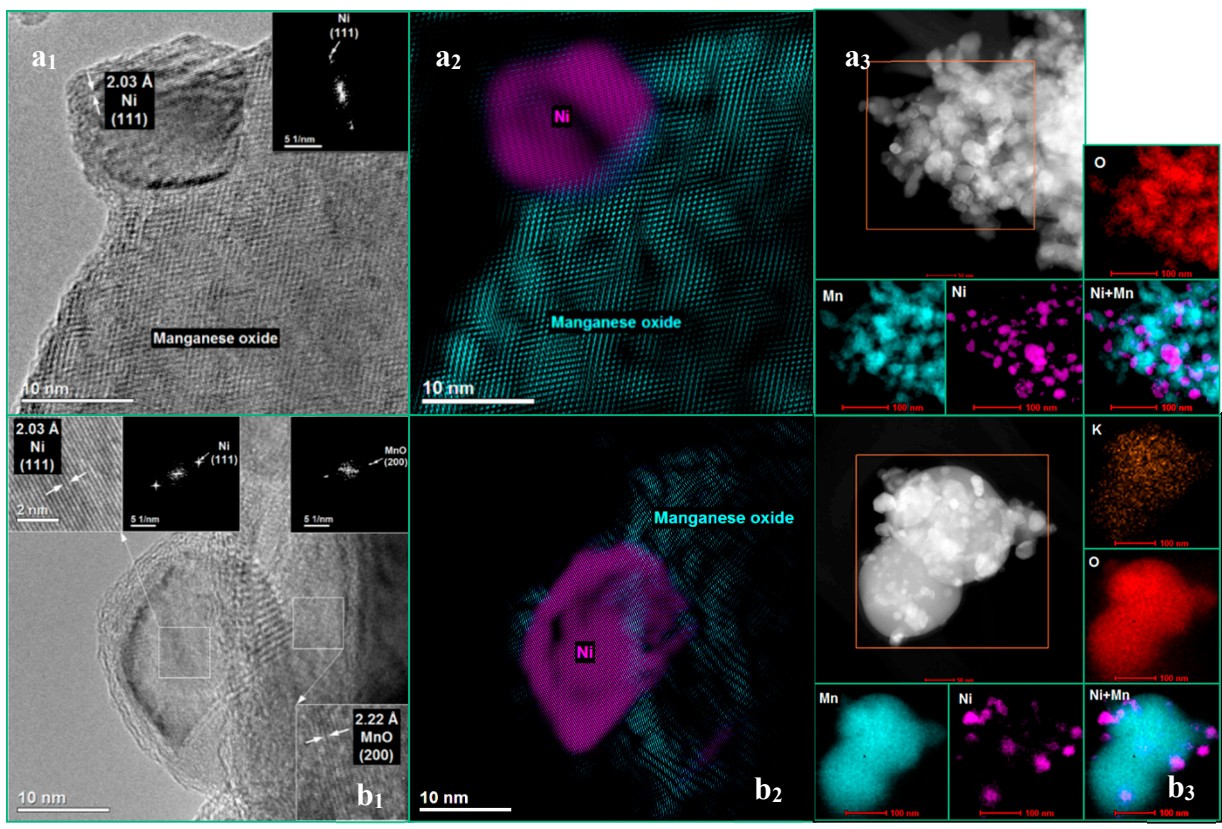

**Figure 1.** Microscopic analysis of the (**a**) Ni/MnO$_x$ and (**b**) KNi/MnO$_x$ catalysts after reduction at 500 °C. HRTEM images (**a$_1$,a$_2$,b$_1$,b$_2$**) and STEM-EDS analysis (**a$_3$,b$_3$**).

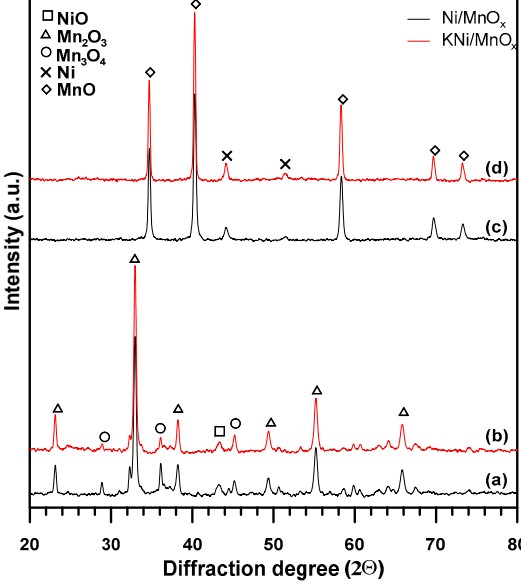

**Figure 2.** XRD patterns of (**a,b**) as prepared and (**c,d**) reduced with hydrogen at 500 °C for Ni/MnO$_x$ and KNi/MnO$_x$ catalysts.

A possible role of the alkali ions as promoters is that they could change the reducibility of the catalyst. Therefore, the relative reducibility of the nickel-based catalysts was investigated using an $H_2$-TPR measurement. Because it is known that NiO and $MnO_x$ oxides are reduced approximately in the same temperature range [43–45] (see Supporting Information, Figure S3), the $H_2$-TPR profile of NiO, KNiO, $MnO_x$, and $KMnO_x$ samples used as the reference samples were also obtained. As seen in Figure 3, the reduction of the $MnO_x$ support can be divided into two stages, namely a low-temperature reduction (LTR) ranging ca. 220–330 °C and a high-temperature reduction (HTR) ranging ca. 330–520 °C. The former reduction peaks can be attributed to the reduction of $Mn_2O_3$ to $Mn_3O_4$, and the later reduction peak can be ascribed to the transformation of $Mn_3O_4$ to MnO, which is in line with the X-ray diffraction result (see Supporting Information, Figure S2) [17,44,46,47]. Compared to $MnO_x$, the reduction peaks of $Ni/MnO_x$ catalysts obviously shifted to lower temperatures, suggesting the excellent redox ability of the sample. The first main reduction peak shifted from 417 °C to 384 °C and the second main reduction peak shifted from 309 °C to 288 °C. The low-temperature peak can be ascribed to the reduction of $Mn_2O_3$ to $Mn_3O_4$, and the high-temperature peak can be ascribed to the overlap of two reduction processes: the reduction of $Mn_3O_4$ to MnO and the reduction of NiO to $Ni^0$. The promotion of $MnO_x$ reduction to MnO by nickel can be interpreted in terms of the activation and spillover of hydrogen from the initially reduced nickel ($Ni^0$) to $MnO_x$ [48,49]. Because the K-containing materials generally show a wide peak and move to a lower temperature [50], the $H_2$-TPR profile of the $KNi/MnO_x$ catalyst shows only one reduction consumption peak between 160–420 °C. The main highly asymmetric peak with the maximum at 350 °C could be attributed to two overlapping processes: rapid reduction of $Mn_2O_3$ to MnO with $Mn_3O_4$ as intermediates and the reduction of NiO to $Ni^0$. A small shoulder peak in the low-temperature region represents the existence of unstable species with different Mn–O bond strengths, which are unstable in the lattice of oxides and could be regarded as surface-active species [50]. The peak of reduction gradually shifted to a lower value after the potassium promoter addition, indicating a weakened interaction between nickel species and the $MnO_x$ support compared to the $Ni/MnO_x$ sample [51]. Moreover, according to Grzybek et al. [30], the presence of potassium facilitates the storage of nitrates on the catalyst's surface. Therefore, the appearance of the intense sharp peaks in the $H_2$-TPR profile of $KNi/MnO_x$ catalyst can result from the presence of a significant amount of nitrate residues accumulated on the catalyst's surface. Probably, the nickel presence increases the nitrate reducibility because hydrogen is dissociated on the reduced nickel ($Ni^0$) and then it spills to the nitrate ions and reduces them [52]. The reduction of nitrates explains the high hydrogen consumption obtained for the $KNi/MnO_x$ sample (see Supporting Information, Table S2).

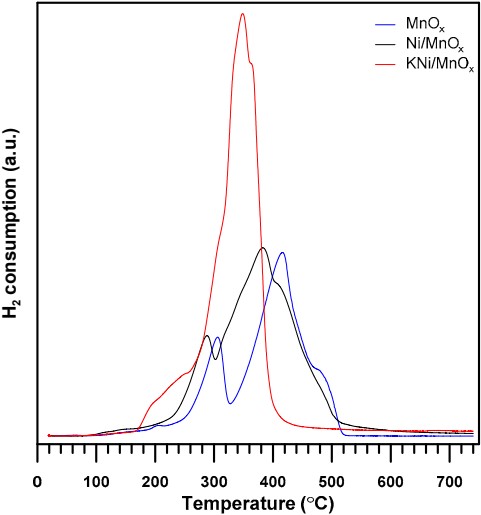

**Figure 3.** $H_2$-TPR profiles of $MnO_x$ support and $Ni/MnO_x$ and $KNi/MnO_x$ catalysts.

In order to better understand the nature of the acid sites and the role of the nickel and potassium addition, in situ FT-IR experiments on $MnO_x$, $Ni/MnO_x$, and $KNi/MnO_x$ samples were performed (Figure 4). For the $MnO_x$ support, the IR spectra of pyridine sorption show the presence of the 1445 cm$^{-1}$ band corresponding to Py interacting with Lewis acid sites with medium strength, which originate from the Mn(IV, III) surface exposed cations. The complexity of the PyL band at 1445 cm$^{-1}$ points to the heterogeneous nature of Mn surface cations ruled by their various oxidation states (IV, III) and/or their coordination with $O^{2-}$. The deposition of nickel resulted in the appearance of an additional band at 1430 cm$^{-1}$, which originated from the physisorbed Py molecules. The total concentration of Lewis acid sites of the $Ni/MnO_x$ catalyst is comparable to that of the $MnO_x$ support (Table 2). However, their strength is higher, giving an intense peak at 1448 cm$^{-1}$. Morrow et al. [53] demonstrated that nitrogen-coordinated pyridine chemisorbs perpendicular to the surface on nickel-metal moieties; thus, such metal-originated Lewis acid sites can give rise to 1448 cm$^{-1}$ IR bands [54]. Consequently, it can be concluded that the presence of metallic forms of nickel does not reduce the number of Lewis acid centers, but furthermore contributes to the increase of the strength of Lewis acid centers.

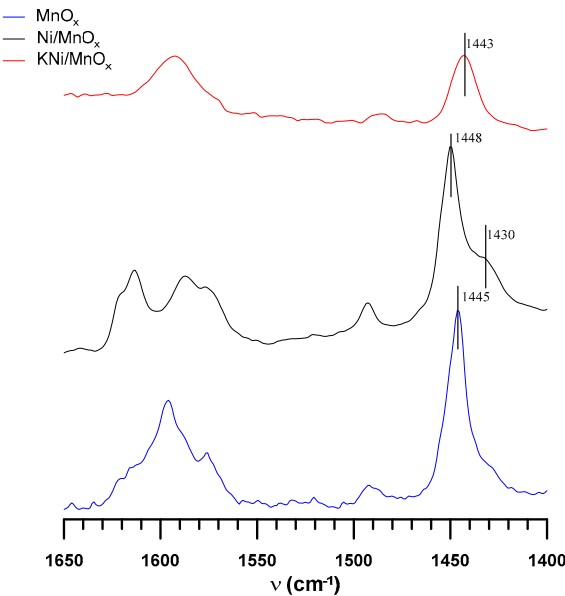

**Figure 4.** FT-IR spectra of pyridyne interacting with the Lewis acid sites of $MnO_x$ support and $Ni/MnO_x$ and $KNi/MnO_x$ catalysts.

**Table 2.** The comparison of the content of Lewis acid sites for $MnO_x$ support and $Ni/MnO_x$ and $KNi/MnO_x$ catalysts.

| Sample | Lewis Acid Sites Concentration (mmol g$^{-1}$) |
|---|---|
| $MnO_x$ | 54 |
| $Ni/MnO_x$ | 50 |
| $KNi/MnO_x$ | 13 |

The potassium addition to the $Ni/MnO_x$ catalyst resulted in the downshift of this band to 1443 cm$^{-1}$, indicating the decrease of the electron acceptor properties of the Lewis sites, which was additionally accompanied by a significant decrease in the number of Lewis acid centers. This phenomenon is ascribed to an increase in the surface electron density induced by K-dopant. This is in line with the results of the work function studies, which show a decrease in the work function of the $Ni/MnO_x$ catalysts upon potassium doping from 4.9 to 4.6 eV, also in accordance with the previous literature reports [55].

### 3.2. The Influence of a Potassium Promoter on the Performance of the Catalysts in Ethanol Steam Reforming Process

The influence of a potassium promoter on the nickel-based catalysts in the SRE process was evaluated in terms of ethanol conversion (Figure 5) and selectivity to products (Figure 6a,b) at the temperature of 420 °C for different $H_2O/EtOH$ molar ratios of 12/1, 9/1, 6/1, and 4/1. Additionally, the performance of the $MnO_x$ support under SRE conditions was carried out (see Supporting Information, Figure S4) and was found to be almost inactive in the SRE process with ethanol conversion lower than 5%. As expected, due to its basicity, $MnO_x$ favors ethanol dehydrogenation to acetaldehyde [56]. The lack of any C1 compound among the products confirms an inability of $MnO_x$ to break the C–C bond in the absence of the nickel active phase. These results are in line with those obtained for the $MnO_x$ support performance under SRE conditions by Gac et al. [21].

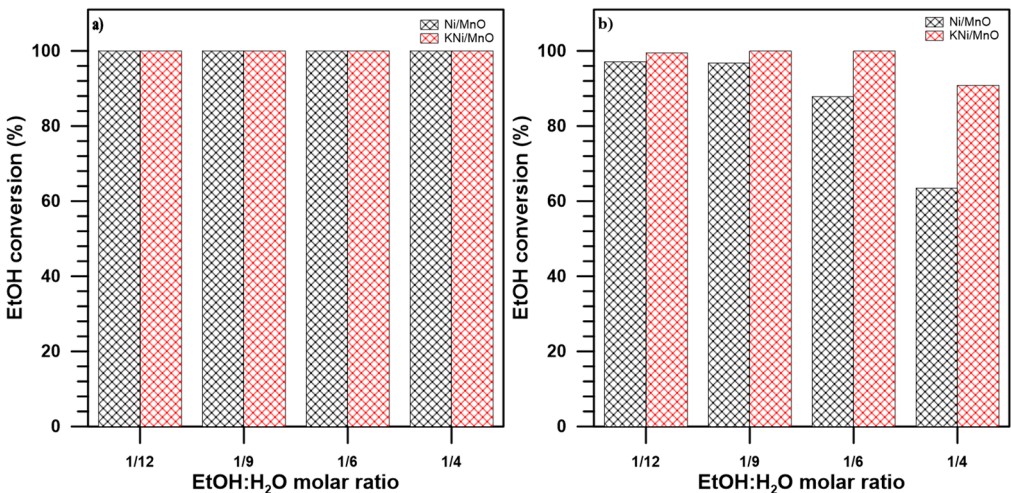

**Figure 5.** The ethanol conversion (**a**) at the beginning and (**b**) after 18 h of SRE process at 420 °C for $H_2O/EtOH$ molar ratio of 12/1, 9/1, 6/1, and 4/1 over $Ni/MnO_x$ and $KNi/MnO_x$ catalysts.

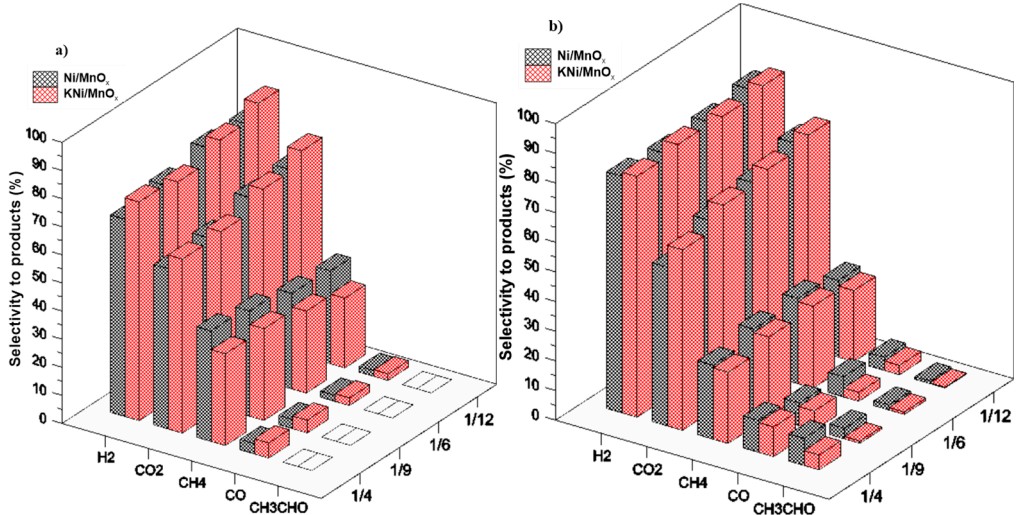

**Figure 6.** The selectivity to products (**a**) at the beginning and (**b**) after 18 h of SRE process at 420 °C for $H_2O/EtOH$ molar ratio of 12/1, 9/1, 6/1, and 4/1 over $Ni/MnO_x$ and $KNi/MnO_x$ catalysts.

Regardless of the $H_2O/EtOH$ molar ratio, at the beginning of the SRE process, both catalysts exhibited 100% ethanol conversion. After 18 h of processing, the ethanol conversion over the $Ni/MnO_x$ sample decreased in the whole range of molar ratios studied because of the carbon formation on the catalyst surface (the studies on catalyst deactivation will be discussed in the next section). The higher excess of water facilitates carbon gasification [56].

Therefore, the highest ethanol conversion of 97% after 18 h of the SRE process was observed for the $H_2O$/EtOH molar ratio of 12/1 and it decreased with the decrease of excess water to 63% for the $H_2O$/EtOH molar ratio of 4/1. For the $KNi/MnO_x$ catalyst, complete ethanol conversion was obtained for 18 h of SRE reaction in the range of $H_2O$/EtOH molar ratio of 6/1–12/1. The decrease of the sample activity to ca. 90% after 18 h of SRE reaction was only observed at the low $H_2O$/EtOH molar ratio of 4/1. The obtained results indicate that the potassium promoter inhibits catalyst deactivations, thus improving its stability under SRE conditions (see Supporting Information, Figure S5), which is consistent with the previous studies on the role of this catalyst promoter in the studied process [25,26,29,34,57,58]. On the other hand, the stability of nickel-based catalysts in the SRE reaction is not always improved by the addition of potassium, which was shown by Słowik et al. [28].

The distribution of products over both nickel-based catalysts indicates that, besides the SRE reaction (Reaction (R1)), ethanol decomposition (Reaction (R2)) and its dehydrogenation (Reaction (R3)) followed by acetaldehyde decomposition (Reaction (R4)) also occur.

$$C_2H_5OH + 3H_2O \leftrightarrow 2CO_2 + 6H_2 \tag{R1}$$

$$C_2H_5OH \leftrightarrow CH_4 + CO + H_2 \tag{R2}$$

$$C_2H_5OH \leftrightarrow CH_3CHO + H_2 \tag{R3}$$

$$CH_3CHO \leftrightarrow CH_4 + CO \tag{R4}$$

At the beginning of the SRE reaction (Figure 6a), regardless of $H_2O$/EtOH molar ratio, both $Ni/MnO_x$ and $KNi/MnO_x$ catalysts exhibit the highest selectivity to hydrogen and carbon dioxide, suggesting that the SRE is the main reaction taking place (Reaction (R1)). More of the most desirable products of SRE reaction is produced over $KNi/MnO_x$ catalyst which could be ascribed to increase the activity of SRE (Reaction (R1)) and the water gas shift (WGS, Reaction (R5)) reactions in the presence of potassium promoted sample.

$$CO + H_2O \leftrightarrow CO_2 + H_2 \tag{R5}$$

Moreover, the rate of both SRE (Reaction (R1)) and WGS (Reaction (R5)) reactions increases with the increase in steam excess [59]. Therefore, in the case of both nickel-based catalysts, the selectivity to carbon dioxide is the lowest for the $H_2O$/EtOH molar ratio of 4/1 (57% for $Ni/MnO_x$, 62% for $KNi/MnO_x$) but increases with its increase to 12/1 (65% for $Ni/MnO_x$ and 73% for $KNi/MnO_x$). At the beginning of the SRE reaction, carbon monoxide and methane are only byproducts that are formed under SRE conditions. In the whole range of $H_2O$/EtOH molar ratios studied, the selectivity to carbon monoxide does not exceed 3.5 and 5% over the $Ni/MnO_x$ and $KNi/MnO_x$ catalysts, respectively. However, selectivity to the second byproduct, methane, is much higher in the presence of both materials because of the high activity of nickel-based catalysts in the methanation reactions of carbon monoxide and/or carbon dioxide (Reactions (R6) and (R7)) [60].

$$CO + 3H_2 \leftrightarrow CH_4 + H_2O \tag{R6}$$

$$CO_2 + 4H_2 \leftrightarrow CH_4 + 2H_2O \tag{R7}$$

The decrease in selectivity to methane from 36 to 33% ($Ni/MnO_x$) and from 32 to 25% ($KNi/MnO_x$) with the increase in the $H_2O$/EtOH molar ratio from 4/1 to 12/1 confirms that excess of water facilitates the steam reforming of methane (Reaction (R8)) [60].

$$CH_4 + H_2O \leftrightarrow CO + 3H_2 \tag{R8}$$

After 18 h of the SRE reaction (Figure 6b), the distribution of products in the presence of both nickel-based catalysts is comparable to that observed at the beginning of the process for a whole range of $H_2O$/EtOH molar ratios studied. The sole appearance of acetaldehyde among byproducts after 18 h of the SRE reaction indicates a decrease in the ability of both

catalysts in the C–C bond cleavage with the increase in time-on-stream due to carbon formation. Regardless of the $H_2O$/EtOH molar ratio, more nickel active sites to break this bond remained available in the case of the potassium-promoted catalyst, resulting in its lower selectivity to the C2 product, i.e., acetaldehyde, and higher selectivity to the C1 products, i.e., carbon monoxide, carbon dioxide, and methane, in comparison with the Ni/MnO$_x$ sample. Because steam excess limits the carbon formation, the amount of produced acetaldehyde increases with a decrease in $H_2O$/EtOH molar ratio.

### 3.3. The Influence of a Potassium Promoter on Prevention of the Nickel-Based Catalyst Deactivation under SRE Conditions

The SEM (Figures 7a$_1$ and 8a$_1$) and TEM images (Figures 7a$_2$,a$_3$ and 8a$_2$,a$_3$) of the Ni/MnO$_x$ and KNi/MnO$_x$ catalysts show that carbon filamentous deposits are formed on the surface of both samples after 18 h of the SRE reaction at 420 °C, regardless of the $H_2O$/EtOH molar ratio. Carbon filaments, leading to catalyst deactivation, are typically formed at the surface of nickel particles under SRE conditions via the Boudouard reaction (Reaction (R9)) and/or methane decomposition [51].

$$2CO \leftrightarrow C + CO_2 \tag{R9}$$

$$CH_4 \leftrightarrow C + 2H_2 \tag{R10}$$

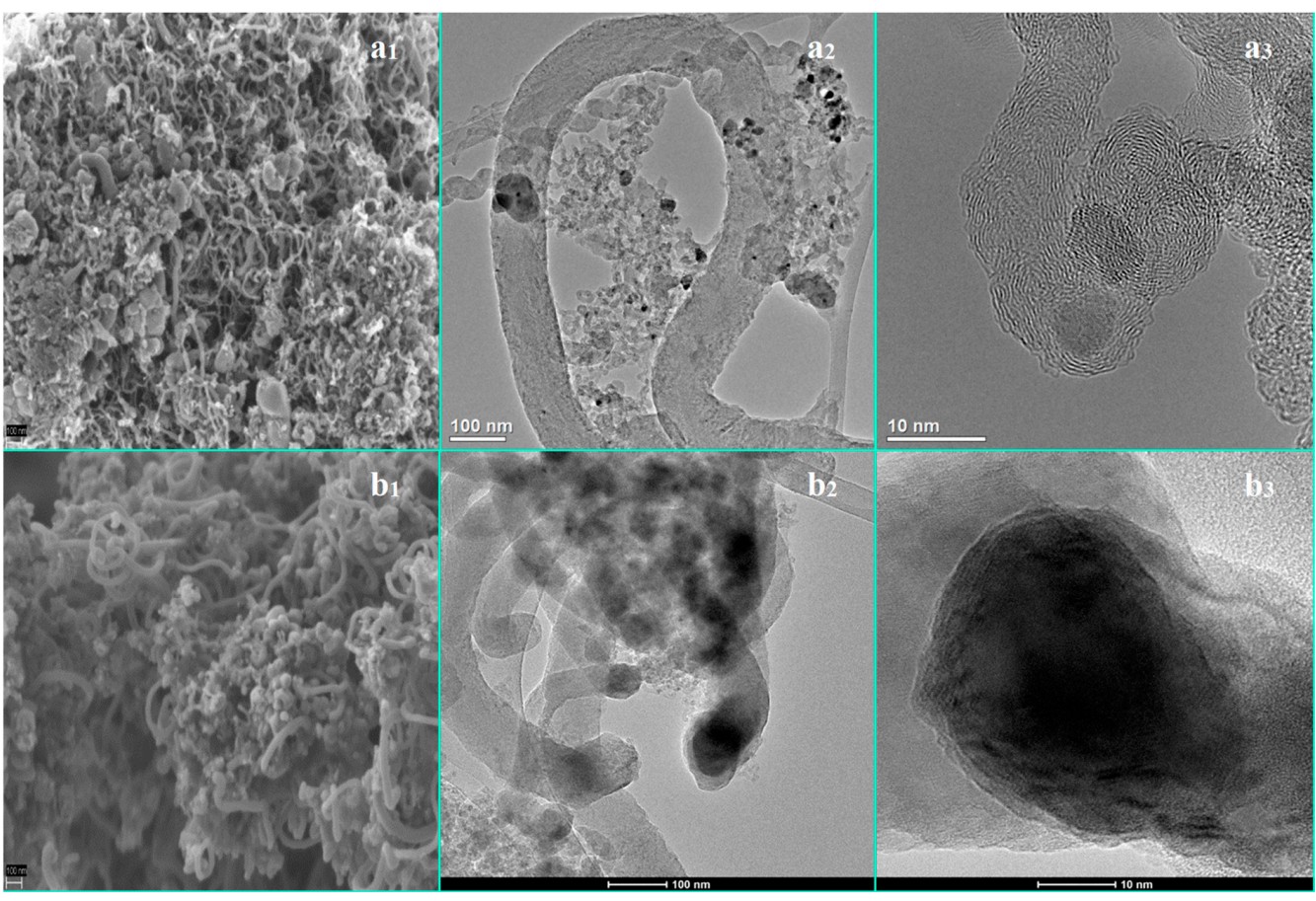

**Figure 7.** Microscopic analysis of Ni/MnO$_x$ catalyst after 18 h of SRE reaction at 420 °C for (**a**) $H_2O$/EtOH molar ratio of 12/1 and (**b**) $H_2O$/EtOH molar ratio of 4/1. SEM images (**a$_1$,b$_1$**) and TEM images (**a$_2$,a$_3$,b$_2$,b$_3$**).

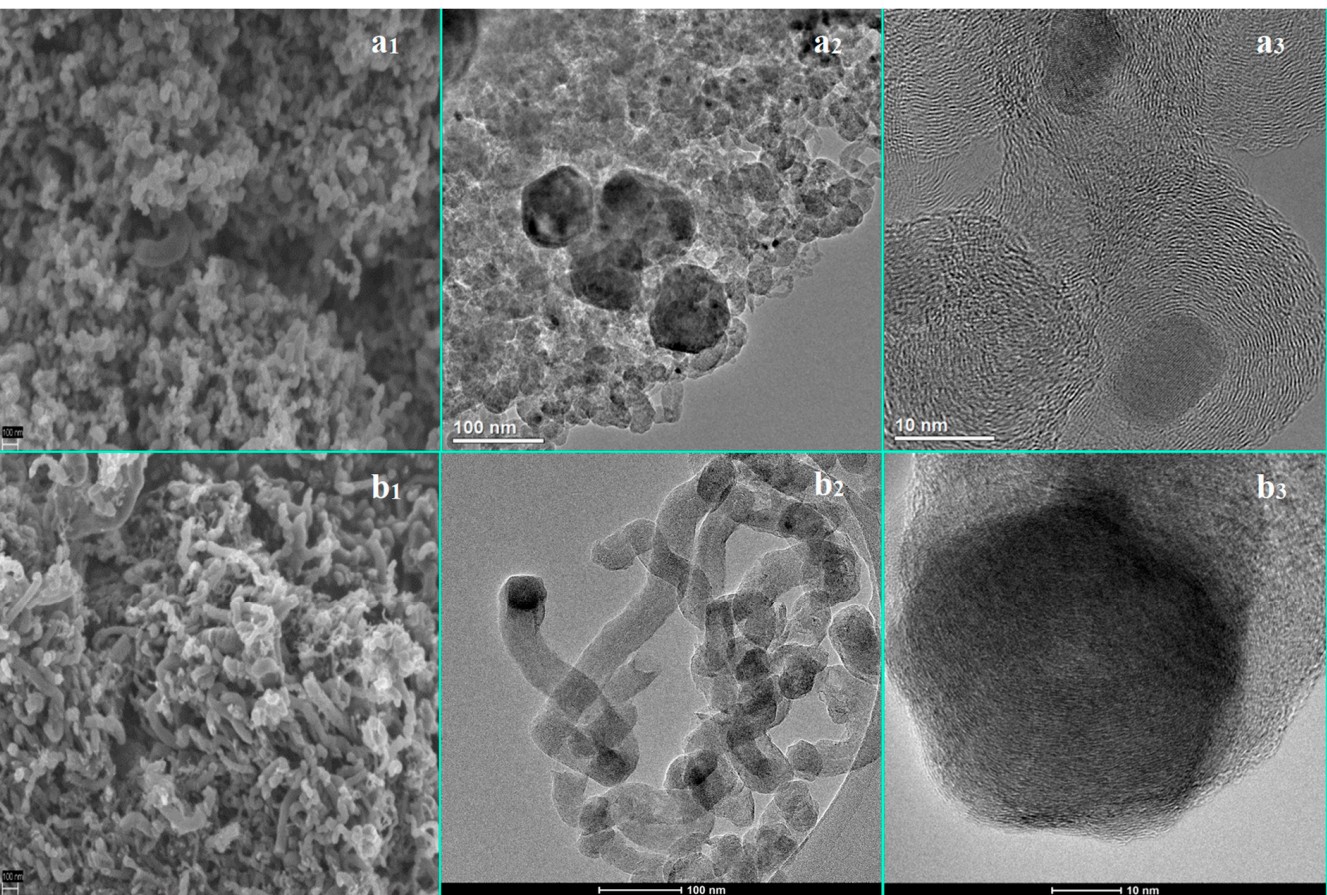

**Figure 8.** Microscopic analysis of $KNi/MnO_x$ catalyst after 18 h of SRE reaction at 420 °C for (**a**) $H_2O/EtOH$ molar ratio of 12/1 and (**b**) $H_2O/EtOH$ molar ratio of 4/1. SEM images (**$a_1$,$b_1$**) and TEM images (**$a_2$,$a_3$,$b_2$,$b_3$**).

Therefore, comparing SEM images of spent catalysts and support (see Supporting Information, Figure S6) reveals that carbon deposition occurs efficiently on nickel surface due to its ability to break C–C and C–H bonds, while relatively little (or lack) carbon deposition occurs on $MnO_x$ without nickel. Since the size distribution of the nickel particles was not in a narrow range for both $Ni/MnO_x$ and $KNi/MnO_x$ catalysts (see Supporting Information, Figure S1), the carbon filaments exhibited a correspondingly non-uniform diameter distribution. Moreover, carbon filaments with not only significantly varying diameters but also lengths formed on the surface of both nickel-based catalysts under SRE conditions. The presence of potassium seems not to influence both diameters and lengths of the carbon filaments. However, the degree of carbon graphitization depends on the potassium promotion, as can be seen in the TEM images (Figures 7$a_3$ and 8$a_3$). The presence of potassium influenced the increase in the degree of graphitization and a highly ordered, i.e., graphitic carbon formed on the surface of the $KNi/MnO_x$ catalyst, whereas mixed turbostratic carbon and graphitic carbon phases formed on the surface of the $Ni/MnO_x$ sample. These results are consistent with those obtained by Słowik et al. [28], who observed an increase in the degree of graphitization of carbon formed under SRE conditions over $KNi/CeO_2$ compared to the $Ni/CeO_2$ catalyst.

By comparing spatial features in Ni and Mn maps of both catalysts after the SRE reaction (Figures 9$a_5$ and 10$a_5$), it is apparent that there are regions with the presence of Ni but without Mn copresence. This means that the Ni particles did not remain attached to the support surface but a carbon deposit separated the Ni phase from the support. Upon closer examination (Figures 7$a_3$,$b_3$, 8$a_3$,$b_3$, 9$a_1$–$a_5$ and 10$a_1$–$a_5$), it was evident that the Ni particles were carried away from the support surface by the carbon growth process

during the reaction and they were located at the tip of the filaments, which is a direct consequence of the weak metal-support interaction(s) associated with both $Ni/MnO_x$ and $KNi/MnO_x$ catalysts. In contrast to these results, where a potassium promotion does not enhance the interaction between Ni and $MnO_x$ phases, the strong cobalt-support interaction resulting from the potassium presence resulted in the stabilization of the active phase, preventing its removal from the $Al_2O_3$ surface because of carbon filament growth under SRE conditions [30].

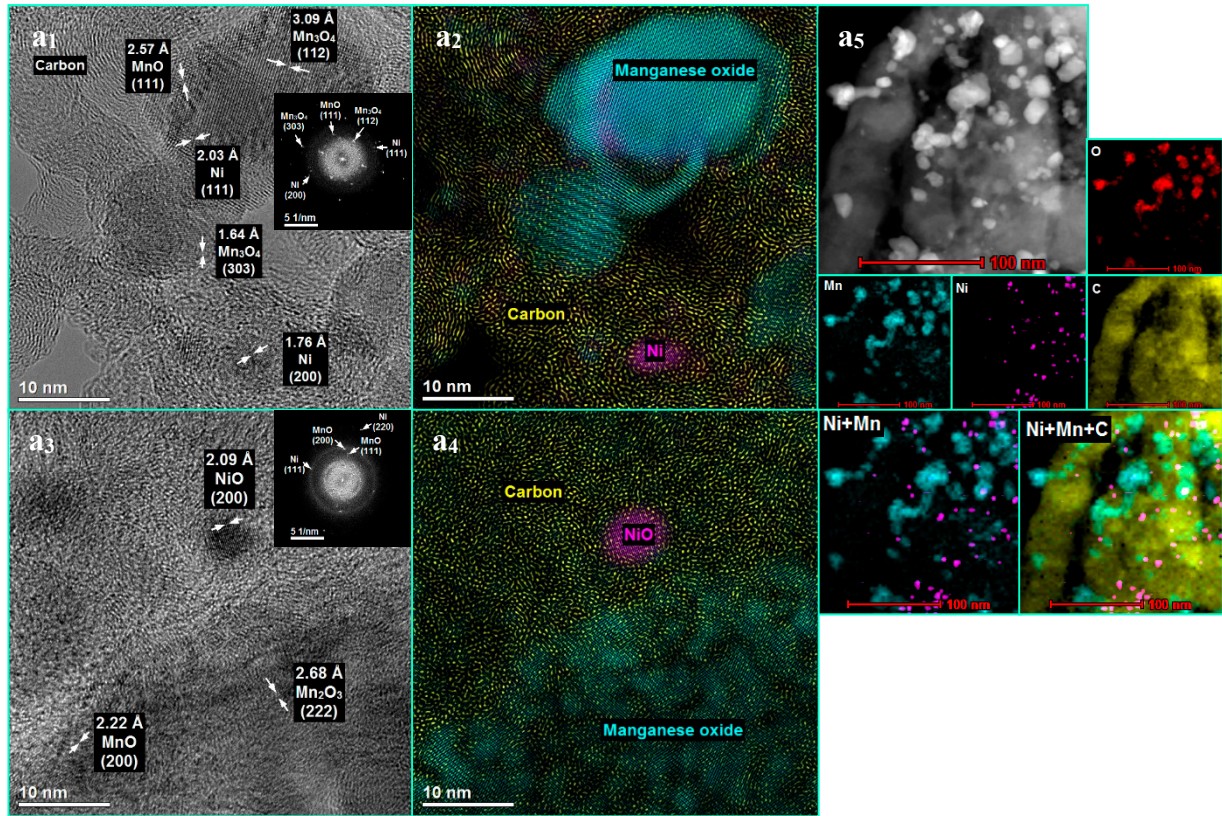

**Figure 9.** Microscopic analysis of the $Ni/MnO_x$ catalyst after 18 h of SRE reaction at 420 °C ($H_2O$/EtOH molar ratio = 12/1). HRTEM images ($a_1$,$a_2$,$a_3$,$a_4$) and STEM-EDS analysis ($a_5$).

The TEM analysis of both nickel-based samples after the SRE process at the $H_2O$:EtOH molar ratio of 12/1 indicates modifications in the distribution of nickel particle size compared to results obtained for the catalysts after their reduction (see Supporting Information, Figures S1 and S7). A comparison of histograms of the reduced samples with those obtained for catalysts after the SRE reaction shows that the larger crystallites mostly disappeared. The average nickel particle size underwent ~50% reduction (Table 3) after the SRE process at the $H_2O$:EtOH molar ratio of 12/1. This suggests that the fragmentation of the initial metal surface occurs prior to the growth of carbon filaments. On the other hand, the nickel particle size distribution obtained for nickel-based catalysts after the SRE reaction at the $H_2O$:EtOH molar ratio of 4/1 was broader and shifted to larger particles (5–65 nm) (see Supporting Information, Figure S8) in comparison with results obtained for them after reduction (see Supporting Information, Figure S1). The increase of particle size after the SRE reaction at the $H_2O$:EtOH molar ratio of 4/1 results from sintering. However, the number of large crystallites is not high and the average particle size (Table 2) is comparable with those obtained for reduced catalysts. The mainly small crystallites were observed after the SRE reaction at the $H_2O$:EtOH molar ratio of 4/1, suggesting that nickel crystallites underwent a continued fragmentation into smaller ones.

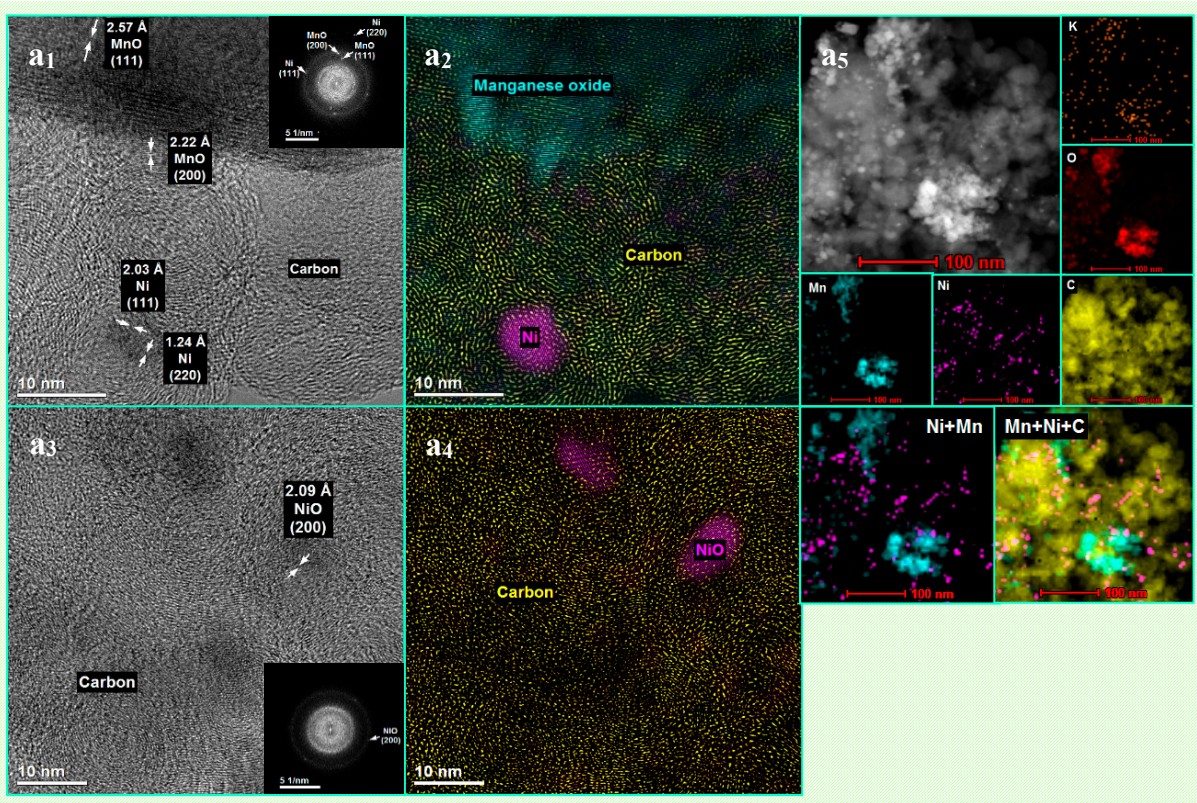

**Figure 10.** Microscopic analysis of the KNi/MnO$_x$ catalyst after 18 h of SRE reaction at 420 °C (H$_2$O/EtOH molar ratio = 12/1). HRTEM images (**a$_1$,a$_2$,a$_3$,a$_4$**) and STEM-EDS analysis (**a$_5$**).

**Table 3.** Ni particle size of the Ni/MnO$_x$ and KNi/MnO$_x$ catalysts after SRE reaction.

| Sample | Ni Particle Size (nm) | |
|---|---|---|
| | H$_2$O:EtOH = 12/1 | H$_2$O:EtOH = 4/1 |
| Ni/MnO | 8 | 18 |
| KNi/MnO | 10 | 20 |

The FFT method (Figures 9a$_1$,a$_3$ and 10a$_1$,a$_3$) reveals the presence of both Ni$^0$ and NiO phases in both spent catalysts. Moreover, based on the phase identification by the FFT method, the MnO$_x$ support of the spent Ni/MnO$_x$ sample was identified to include three forms: MnO, Mn$_2$O$_3$, and Mn$_3$O$_4$, where only facets originating from MnO were detected for the support of the spent KNi/MnO$_x$ catalyst. Moreover, the surface composition was investigated by XPS spectroscopy. The surface of both spent nickel-based catalysts was completely covered by carbon (96 at.%) and oxygen to carbon-bounded species (3.9 at.%); therefore, the intensity of the Mn 2p and Ni 2p regions was very low, almost at the level of noise (Figure 11). Due to this fact, the fitting procedure and calculation of each component's contribution to the overall spectrum were affected by a significant error and will not be discussed in this work. However, it is worth seeing that, on the surface of both catalysts, two oxidation states of manganese and nickel are recorded. The Mn 2p region is slightly shifted towards lower binding energies, suggesting that most manganese may exist as MnO [61]. It should be highlighted that the Mn 2p region overlaps with Ni LMM Auger peaks [62]. In order to analyze the contribution of Ni LMM Auger in overall Mn 2p, we recorded Ni LMM template from Ni0 and NiO. Using these lines and the calculated area ratio between Ni 2p3/2/Ni LMM for Ni0 and Ni 2p 3/2/Ni LMM for NiO, we estimated the contribution of Ni LMM in Mn 2p3/2. Since fitted signals have very small intensity, they were omitted in further analysis of this region. In the case of nickel, the shape of the recorded spectra undoubtedly indicates the presence of NiO (pronounced shoulder at

855.2 eV), whereas the peak at 852.5 eV is related to the metallic species [61,63]. The results obtained from both FFT and XPS methods indicate that both the pre-reduced $Ni^0$ active phase and MnO support of the $Ni/MnO_x$ and $KNi/MnO_x$ samples are oxidized under SRE conditions. However, neither of these methods allow us to determine whether potassium influences the oxidation state of the surface during this process.

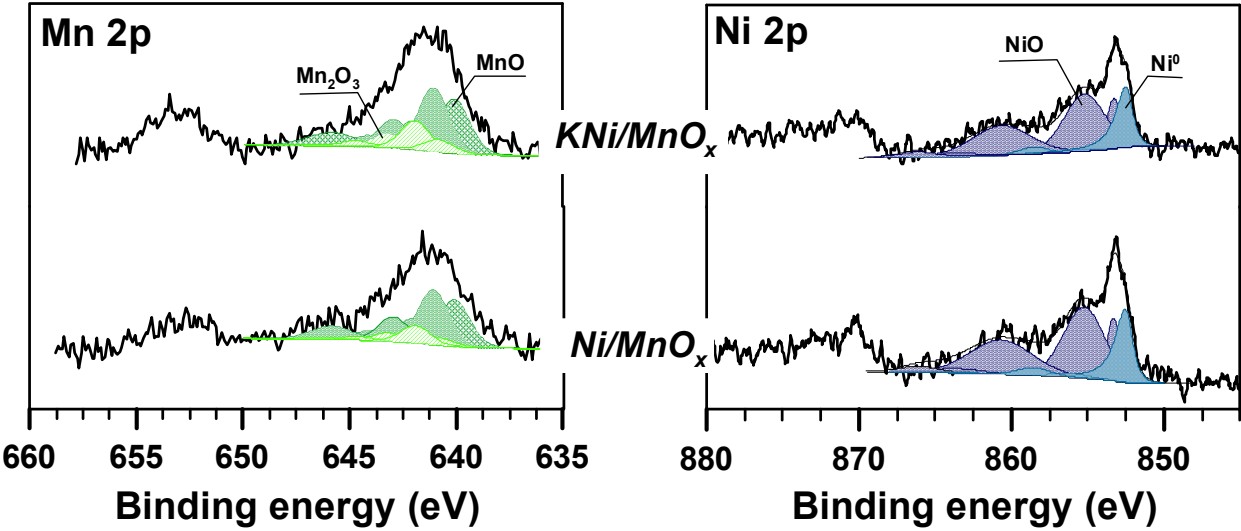

**Figure 11.** High-resolution XPS spectra of Mn 2p and Ni 2p regions collected from the surface of $Ni/MnO_x$ and $KNi/MnO_x$ catalysts after 18 h of SRE reaction at 420 °C ($H_2O$/EtOH molar ratio = 12/1).

To study the difference in the number of accumulated species on the $Ni/MnO_x$ and $KNi/MnO_x$ catalysts and the rate of their formation, the thermogravimetric experiments under SRE conditions were conducted for the $H_2O$/EtOH molar ratio of 12/1 and 4/1 at 420 °C (Figure 12). The results from the thermogravimetric experiments show that the amount of carbon deposits for both nickel-based catalysts is minimized by performing the SRE reaction with the higher excess of water. Moreover, regardless of the $H_2O$/EtOH molar ratio, the rate of carbon formation is significantly lower over $KNi/MnO_x$ compared to $Ni/MnO_x$, which indicates that the addition of an alkali promoter inhibits carbon deposition and/or promotes its gasification. According to the literature [64–66], the presence of potassium on the Ni surface can decrease the number of active sites available for methane decomposition (reaction 10), which allows for suppression of the amount of carbon deposits. The presence of potassium dopant promotes the adsorption of water in a dissociative manner [65,66] and raises the oxygen population on the surface, which, in turn, increases the rate of the steam gasification of carbon formed during steam reforming reactions. Furthermore, the lower accumulation of the carbon deposits on the surface of the $KNi/MnO_x$ catalyst guarantees more Ni active sites available to the reactants and the higher stability of this catalyst compared to the $Ni/MnO_x$ sample.

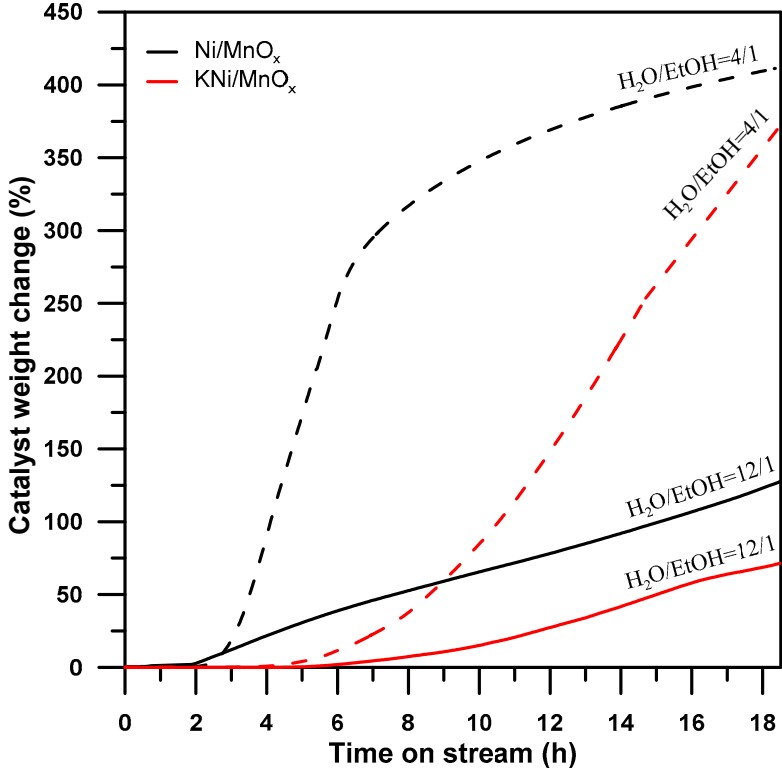

**Figure 12.** Changes in $Ni/MnO_x$ and $KNi/MnO_x$ catalysts' weight under SRE conditions at $H_2O/EtOH$ molar ratio = 12/1 (straight line) and $H_2O/EtOH$ molar ratio = 4/1 (dashed line) at 420 °C.

## 4. Conclusions

By modifying $Ni/MnO_x$ with a potassium promoter, the catalyst stability can be improved under the SRE conditions for different $H_2O/EtOH$ molar ratios in the range from 4/1 to 12/1. Moreover, since the $KNi/MnO_x$ catalyst is more resistant to deactivation, more nickel active sites to break the C–C bond remain available on its surface, resulting in its lower selectivity to acetaldehyde and higher selectivity to C1 products in comparison to the $Ni/MnO_x$ sample after 18 h of the SRE reaction regardless of the $H_2O/EtOH$ molar ratio. The observed high stability of the catalyst promoted with the potassium results from an improved resistance to carbon formation. The potassium promoter inhibits the accumulation of the carbon on the catalyst surface under SRE conditions. Because the Ni active phase dispersion over the $KNi/MnO_x$ sample is worse compared to the $Ni/MnO_x$ catalyst, the observed higher ability to inhibit the carbon formation of potassium-promoted catalysts does not result from the smaller size of Ni crystallites. It means that the restriction of the amount of carbon deposits originates from the potassium presence on the Ni surface, leading (i) to a decrease in the number of active sites available for methane decomposition and (ii) to an increase in the rate of the steam gasification of carbon formed during SRE reactions. On the other hand, the potassium addition does not influence the type of carbon deposits and filamentous carbon is formed in the presence of both $Ni/MnO_x$ and $KNi/MnO_x$ catalysts. Additionally, the morphology, i.e., thickness and length, of carbon filaments formed on the surface of both nickel-based catalysts are similar. Only the degree of graphitization of the carbon deposit on the surface of the $Ni/MnO_x$ and $KNi/MnO_x$ catalysts is slightly different. The $Ni/MnO_x$ catalyst exhibits less graphitization degree than carbon deposited on the surface of $KNi/MnO_x$ material.

**Supplementary Materials:** The following supporting information can be downloaded at: https://www.mdpi.com/article/10.3390/catal12060600/s1, Figure S1: Particle size distribution histograms for (a) $Ni/MnO_x$ and (b) $KNi/MnO_x$ catalysts reduced at 500 °C. Size distribution of particles determined from measurement of at least 200 particles from representative HRTEM images; Figure S2: XRD patterns of (a) calcined and (b) reduced at 500 °C of $MnO_x$ support; Figure S3: $H_2$-TPR profiles of NiO and K/NiO samples; Figure S4: Ethanol conversion and selectivity to products over $MnO_x$ support in SRE process at 420 °C. ($H_2O$/EtOH = 12/1); Figure S5: Stability tests of $Ni/MnO_x$ and $KNi/MnO_x$ catalysts at temperature of 420 °C under SRE conditions for $H_2O$/EtOH molar ratio of (a) 12/1 (b) 9/1, (c) 6/1 and (d) 4/1; Figure S6: SEM images of $MnO_x$ support (a) in the fresh state and (b) after 18 hours of SRE reaction at 420 °C ($H_2O$/EtOH molar ratio = 12/1); Figure S7: Particle size distribution histograms for (a) $Ni/MnO_x$ and (b) $KNi/MnO_x$ catalysts after SRE reaction at temperature of 420 °C under SRE conditions for $H_2O$/EtOH molar ratio of 12/1. Size distribution of particles determined from measurement of at least 200 particles from representative HRTEM images; Figure S8: Particle size distribution histograms for (a) $Ni/MnO_x$ and (b) $KNi/MnO_x$ catalysts after SRE reaction at temperature of 420 °C under SRE conditions for $H_2O$/EtOH molar ratio of 4/1. Size distribution of particles determined from measurement of at least 200 particles from representative HRTEM images; Table S1: XRD details with 2θ, and hkl values of the obtained crystalline phases for as-prepared and reduced with hydrogen at 500 °C of $Ni/MnO_x$ and $KNi/MnO_x$ catalysts [37–42]; Table S2: Hydrogen consumption from $H_2$-TPR results.

**Author Contributions:** Conceptualization, M.G.; Formal analysis, M.G.; Funding acquisition, M.G.; Investigation, M.G., M.R., G.S., S.T.-S., G.G. and K.G.-M.; Methodology, M.G., M.R., G.S., S.T.-S., G.G., K.G.-M. and A.K.; Writing—original draft, M.G. All authors have read and agreed to the published version of the manuscript.

**Funding:** This work was supported by the National Science Centre, Poland within the SONATA program under grant no. 2015/19/D/ST5/01931.

**Acknowledgments:** The study was carried out with equipment purchased thanks to the financial support of the European Regional Development Fund in the framework of the Polish Innovation Economy Operational Program (contract no. POIG.02.01.00-06-024/09 Centre of Functional Nanomaterials; www.cnf.umcs.lublin.pl accessed on 29 April 2022). The SEM and XPS studies of catalysts used in SRE process were carried out at the Biological and Chemical Research Centre, University of Warsaw, established within the project co-financed by the European Union from the European Regional Development Fund under the Operational Programme Innovative Economy, 2007–2013 and by the National Center for Research and Development within the Panda 2 program (contract no.501-D312-56-0000002).

**Conflicts of Interest:** The authors declare no conflict of interest.

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
