# Peer review of "Effect of Potassium Promoter on the Performance of Nickel-Based Catalysts Supported on MnOx in Steam Reforming of Ethanol"

_catalysts, doi:10.3390/catal12060600_

Round 1
Reviewer 1 Report
Ms. ID.: catalysts-1728409
Title: Production of hydrogen from steam reforming of ethanol nickel based catalysts supported on MnOx: The influence of potassium as a promoter
This manuscript presents an experimental work focused on hydrogen production by steam reforming of ethanol. The influence of potassium in the nickel based catalyst supported on MnOx has been studied. The catalysts have been characterized by several techniques and the results show a better stability of the catalyst promoted with potassium. This manuscript is suitable for publishing after minor revisions.
Comments:
1.- In section 2.1 Catalysts synthesis, the authors should provide more details about catalyst preparation by impregnation. The amount of MnOx solid and the amount of the solution of nickel nitrate. Was the preparation by wet impregnation or by incipient wetness impregnation? The same for the impregnation of potassium.
2.- On page 3 lines 124 and 131 appears cobalt, but is must be a typographical error.
3.- The electron-donor properties have been evaluated. The authors should explain in more detail this technique and the results obtained.
4.- In section 2.3 Catalyst evaluation in SRE process, the authors should indicate the compounds analyzed in each chromatograph, Bruker 450-GC and Bruker 430-GC. It should be explained if EtOH was also determined by chromatography. The authors should explain why the conversion of ethanol was calculated from the concentrations before and after the reaction. I consider to be better using moles before and after the reaction.
5.- Correct X-axis in Figure 4. Equations (5) to (10) are equilibrium. Correct equation (6). Correct the number of figures after Figure 5, also in the text.
6.- The authors should better explain the sentence: “Regardless of H2O/EtOH molar ratio, more nickel active sites to break this bond remain available in the case of potassium promoted catalyst resulting in its lower selectivity to acetaldehyde and higher C1 products in comparison with Ni/MnOx sample”. (page 11 lines 415-417). Please specify for the C1 products, CO2, CH4 and CO.
7.- In the Figure numbered as 8, it is not clear a1, a2, ,b1, b2…, please complete.
8.- Correct the legend in the Figure numbered as 12.
Reviewer 2 Report
This manuscript reports on studies of the effect of potassium promoter on the stability and resistance to carbon deposits formation of Ni/MnOx catalyst under SRE conditions. The quality of this manuscript is relatively poor, and there are many major problems. Hence, this paper cannot be suggested to be published in this journal. The detailed problems are given as follows:
- For abstract, according to the characterization results in the article, addition amount of potassium should be the actual weight percentage of loading (1.6%) rather than 2%.
- Include information on chemicals purity used in the Catalysts synthesis section.
- The authors claimed that it is difficult to establish which nickel-based catalyst exhibited more uniformly distributed Ni0 particles, which means that potassium does not influence on the nickel active phase dispersion. But according to Table 1, the KNi/MnOx sample exhibits larger Ni0 particle size and rather low dispersion.
- The authors claimed that there is not much difference between the XRD patterns of Ni/MnOx and KNi/MnOx catalysts, it suggests that no phase is changed after the potassium addition. Actually, there are two obvious peaks in the range of 30-35 ° of KNi/MnOx catalysts, which should be further analyzed.
- The appearance of the intense sharp peaks in the H2-TPR profile of KNi/MnOx catalyst can result from the presence of a significant amount of nitrate residues accumulated in the catalysts surface. The authors did not discuss the possible contributions of the nitrate residues accumulated in the catalytic performance. The authors should also provide the results without the nitrate residues accumulated. Control experiments were not performed in this study. This is not acceptable.
- The authors claimed that FT-IR peak strength of the Ni/MnOx catalyst is higher giving an intense peak at 1448 cm-1, but marked 1445 cm-1 in the figure. The authors also claimed that the potassium addition to the Ni/MnOx catalyst results in the downshift of this band to 1445 cm-1, but marked 1443 cm-1 in the figure.
- The authors claimed that the degree of carbon graphitization depends on the potassium promotion as can be seen in TEM (Fig. 8a6, b6). The presence of potassium influence on the increase in the degree of graphitization and highly ordered graphitic carbon is formed on the surface KNi/MnOx catalyst whereas mixed turbostratic carbon and graphitic carbon phases are formed on the surface of Ni/MnOx sample. However, the potassium addition did not influence on the type and morphology of carbon deposits and filamentous carbon is formed in the presence of both Ni/MnOx and KNi/MnOx catalysts.
- For Section 3.1, the influence of potassium promoter on the catalysts physicochemical properties should be discussed around improved stability. Actually, many characterization results lack the evidence to support the influence of potassium promoter on the improved stability.
- Actually, the KNi/MnOx catalyst with very low BET surface area, rather low nickel dispersion and larger average particle size. In my view, this is not a suitable catalyst for this reaction.
- Figure 12 corresponds the weight change of Ni/MnOx and KNi/MnOx catalysts, rather than Ni/CeO2 and KNi/CeO2 catalysts!
- The authors claimed that the rate of carbon formation is significantly lower over the KNi/MnOx compared to Ni/MnOx, but there are no adequate characterizations for certifying conclusions that "the addition of alkali promoter inhibit carbon deposition and/or promote its gasification. " and "The potassium presence on the Ni surface leading (i) to decrease in the number of active sites available for methane decomposition and (ii) to increase in the rate the steam gasification of carbon formed during SRE reactions."
- More characterization of the spent catalysts should be provided such as specific surface area and pore properties. Did the Ni particle size and distribution change before and after the reaction? The catalyst stability test runs for 18 hours. How about the changes in the catalyst properties after the reaction for 18 h? If possible, authors compare their results of the stability test with previously published articles.
Reviewer 3 Report
Overall the experiments and results are discussed properly, but more insight into the catalyst activity is needed. Here are my detailed comments:
- In the synthesis method, please provide the concentration and the amount (g or mol) of reagents and precursors used.
- Please provide the linear region used for the BET measurement.
- During the catalyst synthesis, initially, nickel was impregnated followed by potassium. Hence the chances of potassium affecting nickel dispersion seem low. What would happen if the potassium was impregnated first followed by the nickel precursor impregnation?
- According to the characterization results, K is highly dispersed. So, K might be blocking the lewis acid sites on MnOx. Could this be the reason for the decrease in Lewis acid sites in KNi/MnOx, instead of K effect on the MnOx electron acceptor properties?
- It will look better if Fig. 5 is split into two separate figures, one figure showing initial activity results and the other showing the activity after 18 h. Please try to improve the figures for activity test results.
- In Fig 8, please label a1-a4 and b1-b4.
- The activity tests were done at high temperatures where the conversions are close to 100 % for both KNi and Ni catalysts, please provide the results at low conversions. Can potassium cover the Ni metal and block the active sites and prevent the formation of carbon deposits? A temperature-dependent study with 1/12 (EtOH/H2O) will show how these two catalysts behave at low conversions. CO uptake values on both the KNi and Ni catalysts will provide information about the availability of Ni active sites. The CO uptake value should then be used to calculate the TOF for temperature-dependent results, this TOF results will provide more evidence on whether K is enhancing the catalyst activity or not. Also use the TOF rates to calculate the activation energies, to see if the active sites have changed or not.
- The microscopic analysis shows that Ni is moved from the MnOx support and is now supported on carbon. Then, what role is the role of MnOx in the reaction? Will we get a similar result with KNi and Ni supported on SiO2?
- “Production of hydrogen from steam reforming of ethanol nickel based catalysts supported on MnOx: The influence of potassium as a promote” The catalyst is not ethanol based, please rewrite it with something like “…over nickel-based catalysts…”.
Round 2
Reviewer 2 Report
1. There are two obvious peaks in the range of 30-35 ° of KNi/MnOx catalysts. The authors claimed that the XRD analysis was repeated and these two reflexes do not appear in the new XRD diffractogram, which suggests sample contamination during the first XRD analysis. However, compared with the original manuscript, it is obvious that the author only modified the original data rather than repeated the XRD experiments.
2. The authors claimed that because the influence of potassium precursor on the catalyst performance in the SRE reaction is a very broad issue, it will be entirely subject to our next paper. The authors should briefly clarify the influence trend and reasons of nitrate residues on catalytic performance, or provide relevant literature at least, which is an essential factor to improve the catalytic performance in this paper.
3. For the rewritten conclusion section, the authors claimed that as inferred from the FFT analysis of the TEM images, the Ni/MnOx catalyst exhibits less graphitization degree than carbon deposited on the surface of the KNi/MnOx material. However, no relative data is provided in Figure 9 and Figure 10, therefore I suggest that it should be added.
4. The authors claimed that this amount of carbon was formed on the surface is several times greater than the amount of the catalysts (100mg) used for the test. If so, this means that the large amount of carbon deposit generated will lead to rapid deactivation of the catalyst or blockage of the quartz tube.
5. The average nickel particle size underwent ~50% reduction (Table 2) after the SRE process at H2O:EtOH molar ratio of 12/1. It suggests that the fragmentation of the initial metal surface occurs prior to the growth of carbon filaments. Unfortunately, there is no evidence to support this claim.
6. The turnover frequency (TOF) must be provided for comparison.
7. Control experiments were not performed and were completely ignored in this study. The authors should also provide the results without catalyst, with catalyst and with quartz.
Reviewer 3 Report
The paper can be accepted in its present form